# Hypothesis: Why Different Types of SDH Gene Variants Cause Divergent Tumor Phenotypes

**DOI:** 10.3390/genes13061025

**Published:** 2022-06-07

**Authors:** Jean-Pierre Bayley, Peter Devilee

**Affiliations:** 1Department of Human Genetics, Leiden University Medical Center, 2333 ZA Leiden, The Netherlands; p.devilee@lumc.nl; 2Department of Pathology, Leiden University Medical Center, 2333 ZA Leiden, The Netherlands

**Keywords:** head and neck paraganglioma, pheochromocytoma, succinate dehydrogenase, reactive oxygen species, neuroendocrine tumor

## Abstract

Despite two decades of paraganglioma-pheochromocytoma research, the fundamental question of how the different succinate dehydrogenase (SDH)-related tumor phenotypes are initiated has remained unanswered. Here, we discuss two possible scenarios by which missense (hypomorphic alleles) or truncating (null alleles) SDH gene variants determine clinical phenotype. Dysfunctional SDH is a major source of reactive oxygen species (ROS) but ROS are inhibited by rising succinate levels. In scenario 1, we propose that SDH missense variants disrupt electron flow, causing elevated ROS levels that are toxic in sympathetic PPGL precursor cells but well controlled in oxygen-sensing parasympathetic paraganglion cells. We also suggest that SDHAF2 variants, solely associated with HNPGL, may cause the reversal of succinate dehydrogenase to fumarate reductase, producing very high ROS levels. In scenario 2, we propose a modified succinate threshold model of tumor initiation. Truncating SDH variants cause high succinate accumulation and likely initiate tumorigenesis via disruption of 2-oxoglutarate-dependent enzymes in both PPGL and HNPGL precursor tissues. We propose that missense variants (including SDHAF2) cause lower succinate accumulation and thus initiate tumorigenesis only in very metabolically active tissues such as parasympathetic paraganglia, which naturally show very high levels of succinate.

## 1. Introduction

Pheochromocytomas-paragangliomas (PPGL) and head and neck paragangliomas (HNPGL) are neuroendocrine tumors that originate in neural crest-derived cells of the sympathetic and parasympathetic nervous systems, respectively. Pheochromocytomas arise from the adrenal medulla and sympathetic paragangliomas from abdominal or thoracic paraganglia. Parasympathetic paragangliomas occur primarily in the head and neck. The two groups of paraganglia are also functionally and clinically distinct; sympathetic paraganglia are mainly associated with catecholamine production and result in tumors with clinical malignancy. Parasympathetic paraganglia have a CO_2_, oxygen and pH-sensing function and resulting tumors are locally highly morbid but rarely metastasize.

Germline and somatic variants in around 20 genes are currently linked to PPGL/HNPGLs, explaining approximately 40% of all cases [1]. Among the most clinically important genes are those encoding the SDHB and SDHD protein subunits of the tricarboxylic acid (TCA) cycle enzyme, succinate dehydrogenase (SDH) [2]. Succinate dehydrogenase is a heterotetrameric protein consisting of two soluble subunits: the primary catalytic subunit, SDHA, and a redox subunit, SDHB, involved in electron transport. The two membrane-spanning subunits, SDHC and SDHD, anchor the complex in the inner mitochondrial membrane and facilitate electron transfer to the mitochondrial electron transport chain where succinate dehydrogenase is referred to as complex II (CII) or succinate:ubiquinone (oxido)reductase (SQR). As with many other tumor suppressor genes, the disease phenotype associated with SDH variants is only expressed once a second genetic event eliminates the remaining functional wildtype allele. In the case of SDH genes, this is primarily due to so-called loss of heterozygosity (LOH) events in the tumor that cause partial or entire loss of the wildtype allele or chromosome [3,4], although secondary somatic mutations are occasionally reported.

### 1.1. Genotype–Phenotype Correlations

Until recently, the various classes of variants in the SDHB and SDHD genes were thought to have comparable phenotypes, with the major clinical distinction drawn between the predominantly PPGL and metastasis-related phenotype of SDHB variants and a predominantly non-metastatic HNPGL phenotype associated with SDHD variants [5,6,7]. No explanation for these genotype–phenotype relationships has been offered to date (Figure 1).

The first group to report a link between an SDH variant and a specific phenotype was that of Eamonn Maher in Birmingham/Cambridge. A high-HNPGL/low-PPGL phenotype was associated with a single SDHD missense variant, p.(Pro81Leu) [10,11]. However, the outcome of those studies was influenced by a prominent founder effect in the dataset, obscuring its wider importance. This was followed by a study from Francesca Schiavi and colleagues in Padua who reported that the SDHD variant p.(Tyr114Cys) was also associated with a high-HNPGL/low-PPGL phenotype [12]. This study again focused on a single variant.

In contrast to these studies, we were able to assemble a large, genetically diverse cohort with sufficient power to examine genotype-phenotype correlations more broadly. This allowed us to go beyond the examination of single variants, identifying the first major genotype-phenotype correlations for overall SDHB and SDHD variant classes [8]. Analyzing three large, genetically diverse and differently ascertained patient datasets from the Netherlands, Germany and the UK, including a total of 950 SDH-associated PPGL/HNPGL patients, we first noticed that truncating SDH variants accounted for many more clinical cases compared to missense variants (summary in Figure 2). Further analysis revealed that carriers of truncating SDHD variants have a significantly higher risk for PPGL (*p* < 0.0001), an earlier age of diagnosis (*p* < 0.0001) and a greater risk for PPGL/HNPGL comorbidity compared to carriers of missense variants in SDHD. In a subsequent detailed analysis of SDHB patient data, truncating SDHB variants were also found to be significantly overrepresented amongst cases with PPGL (*p* = 0.003) and malignancies (*p* = 0.016) compared to missense SDHB variants [9].

### 1.2. Mechanisms of Tumorigenesis

The primary mechanisms by which SDH gene variants are thought to initiate tumorigenesis are (1) the accumulation of ‘oncometabolite’ succinate, leading to various downstream effects such as HIF1alpha stabilization, DNA hypermethylation and functional disruption of a range of important enzymes [13,14,15,16,17,18], (2) the generation of high levels of reactive oxygen species (ROS), which act as intercellular signals or interact with and/or damage proteins, lipids and DNA [19,20,21,22,23] or (3) via energetic changes due to alterations in SDH/SQR activity [24]. Research in recent years has focused on succinate as an inhibitor of the large and diverse 2-oxoglutarate-dependent dioxygenase family that includes enzymes such as prolyl hydroxylases, histone demethylases and TET [4,14,15,16,17,25]. Succinate was also recently reported to cause a deficiency of the homologous recombination DNA repair pathway via inhibition of 2-OG-dependent lysine demethylases [26], as well as vulnerability to PARP inhibitors combined with cytotoxic agents or irradiation [27]. We will not speculate on the underlying drivers of downstream tumorigenesis here. Rather, we will argue that certain SDH variants either prevent or cause tumor initiation in specific precursor cell types, building on the premise that specific tissues have higher or lower tolerances to the metabolic disruptions caused by SDH variants.

In an attempt to explain our phenotypic findings in SDHx variant carriers in molecular and tumor biological terms, we propose two novel, partially overlapping, scenarios. Scenario 1 is based on current understanding of ROS generation resulting from SDH dysfunction, while scenario 2 incorporates observations on succinate accumulation and tissue-specific succinate levels.

## 2. The Paraganglioma ROS Model

### Succinate Dehydrogenase and ROS Generation

Analysis of succinate dehydrogenase protein structure in *E. coli* suggested that the succinate-binding dicarboxylate site is linked to the ubiquinone binding site by a series of redox centers [28]. The first is a flavin adenine dinucleotide (FAD) co-factor located in SDHA, which is responsible for stripping electrons from succinate, followed by three ([2Fe-2S], [4Fe-4S] and [3Fe-4S]) iron-sulfur clusters in SDHB which channel electrons across SDHB, finally terminating at a ubiquinone binding site located at the SDHB/SDHC/SDHD interface, where 2H^+^ and 2e^−^ reduce ubiquinone to ubiquinol as part of the mobile ‘Q cycle’ (Figure 3). The resultant quinol pool supports the regeneration of the NAD+ and FAD cofactors, generation of the proton motive force across the mitochondrial inner membrane and subsequent ATP synthesis (Figure 3) [28].

In a series of papers, Ishii and colleagues reported that a *C. elegans* SDHC mutant (mammalian equivalent SDHC p.Val69Glu) oxidizes succinate to fumarate but is unable to transfer electrons to ubiquinone, causing increased ROS due to the leakage of electrons that fail to reach ubiquinone [29,30,31]. In this model, ROS were primarily generated through the action of SDHA FAD, shown by the ability of excess succinate to suppress ROS generation. Further independent work confirmed and extended these findings, suggesting that succinate and ROS interact either synergistically or antagonistically [21,23,32,33,34]; at lower succinate concentrations any increase leads to a rise in ROS levels, but succinate concentrations of >400 µM result in constant occupation of the succinate binding site, inhibiting ROS generation [33]. High concentrations of succinate (>400 µM) have been reported in SDH KD model systems and tumors [14,15]. Together, these and other findings support the idea that the disruption of normal electron flow via FAD may be a source of high ROS levels and this effect is limited by succinate.

At this point it is worth noting that truncating and missense variants affecting SDHB, SDHC and SDHD may result in ROS generation via slightly different mechanisms. In the absence of SDHB, SDHA can still oxidize succinate [21] but likely at a much lower rate compared to assembled complex II, resulting in lower ROS levels compared to SDHB, SDHC and SDHD missense variants, which may also generate superoxide at the ubiquinone binding site [20,35] in addition to the SDHA FAD site. In general, we predict that certain missense variants in all SDH subunits, but especially SDHB, cause virtual or complete loss of complex II assembly-enzyme activity and thus functionally resemble truncating variants [36]. However, we suspect that across the spectrum of missense variant effects, many variants will show significant levels of residual function sufficient to substantially modify clinical phenotype. This will have to be thoroughly investigated for each individual variant, an undertaking already well underway in other fields such as breast cancer [37,38].

In scenario 1, missense SDH variants produce the highest ROS levels (Figure 4), leading to selective ROS-mediated toxicity in vulnerable sympathetic pheochromocytoma–paraganglioma precursor tissues. ROS-mediated toxicity might also play a role in a broader range of tissues, perhaps explaining the tissue selectivity of SDH variants. We predict that sympathetic paraganglia have weaker anti-oxidative defenses and/or weaker protection against ROS-induced apoptosis compared to oxygen-sensing parasympathetic tissues, which likely require tight redox control due to the direct involvement of ROS in oxygen-sensing [39,40]. Both missense and truncating SDH variants initiate tumorigenesis in parasympathetic tissues, whereas missense-driven toxicity and cell death largely confines tumorigenesis in sympathetic tissues to low-ROS truncating (null) variants. In this scenario, ROS primarily function as an inhibitor of tumorigenesis, and missense SDH gene variants therefore act, somewhat counter-intuitively, as anti-oncogenic tumor suppressors in sympathetic tissues.

## 3. The Succinate Threshold Model

### 3.1. Succinate Dehydrogenase and Succinate Accumulation

Early biochemical studies of SDH loss in cell models and patient tumors found that disruption of succinate oxidation led to the accumulation of high levels of succinate [14,15,41,42,43]. These studies were quickly followed by pioneering work from the labs of Jim Maher and Kenneth McCreath. The Maher lab showed that loss of SDHB in yeast leads to succinate accumulation and inhibition of the 2-oxoglutarate (2OG)-dependent enzyme Jlp1, which is involved in sulfur metabolism, and the 2OG-dependent histone demethylase Jhd1 [16]. The McCreath–Devilee labs then showed that knockdown of either SDHD or SDHB in mammalian cells leads to an increase in steady-state levels of both H3K27me3 and H3K36me2, which could be reversed by overexpression of JMJD3 histone demethylase. We also showed that paraganglioma chromaffin (type I) cells are the highly methylated histone component of paragangliomas [17,44].

### 3.2. The Succinate Threshold Model

These studies provide a backdrop for our second proposed explanation of how different SDH genes cause specific tumor phenotypes. In this model of tumor initiation, we propose that a specific level of succinate must be reached before disruption of 2OG-dependent oxygenases is sufficient to initiate tumorigenesis. Studies suggest that the carotid body may have a very high metabolic rate, with 3-fold higher succinate levels than brain tissue and 8-fold higher levels than the adrenal medulla [45,46,47,48].

We hypothesize that truncating variants block succinate oxidation, leading to significantly higher succinate levels compared to missense variants, levels that are sufficiently high to cause tumor initiation in both sympathetic and parasympathetic tissues. Missense variants, on the other hand, cause incomplete disruption of succinate oxidation, resulting in succinate levels insufficient in most cell types to cause disruption of 2-oxoglutarate-dependent enzymes and tumor initiation (Figure 5) in precursor tissues. Only in highly metabolically active cells, such as the parasympathetic paraganglia, will missense variants result in sufficient succinate accumulation to reach the succinate tumorigenic threshold. This effect is likely independent of subsequent succinate accumulation in clinically apparent tumors, but this point should be clarified.

## 4. How Do Individual SDH Variants Influence Phenotype?

### 4.1. Missense Versus Truncating Variants: ROS versus Succinate

As mentioned above, SDHB is broadly associated with PPGL, while SDHD is mainly associated with HNPGL. We argue that these rudimentary associations mask a more subtle interplay of variant-specific and demographic factors. Rather than phenotypic correlations based on a particular SDH gene, we propose that clinical phenotypes differ primarily on the basis of SDH variants that either modify protein function (hypomorphic alleles) or abolish protein function (null alleles). Not all missense variants result in hypomorphic alleles as some may completely abolish protein function. In vitro data on hypomorphic alleles are sparse and most studies to date have focused on simple knockdown/knockout of SDH genes [14,15,49,50]. Nevertheless, we suggest that missense variants/hypomorphic alleles often result in only partial blockade of SDH function, often accompanied by limited succinate accumulation and high ROS generation, whereas truncating alleles completely abolish SDH activity primarily and lead to high succinate levels accompanied by relatively modest ROS generation. For example, SDHC and SDHD missense variants that disrupt the binding of ubiquinone might be expected to show very different phenotypes from missense variants that obstruct membrane insertion or inhibit interaction with SDHB, preventing complex II assembly. This argument is also likely to be valid for certain missense SDHB variants. Another important factor to keep in mind when considering phenotypic variation is that the biological effect of a variant will not be absolute, but subject to significant modulation by patient-specific and/or stochastic factors. We therefore hypothesize that all clinically apparent SDH variants occupy a spectrum from high ROS/low succinate to high succinate/low ROS (Figure 4).

### 4.2. Tumor-Tissue Specificity

How do these proposed biochemical differences between SDH variant types translate to differences in tumor associations? Based on current knowledge, any model of SDH-driven tumor initiation must start from the premise that ROS, succinate and/or energetic/metabolic changes initiate tumorigenesis. To explain our recent genotype–phenotype findings, any proposal must also explain why the biochemical consequences of missense versus truncating SDH variants would lead to different outcomes in sympathetic paraganglia but similar outcomes in parasympathetic tissues. We propose that SDHB and SDHD missense variants differ in the extent to which they act as hypomorphic versus null alleles. SDHB shows higher evolutionary conservation compared to SDHC/SDHD and has major structural interactions with both SDHA and the membrane-bound SDHC/SDHD. SDHB immunohistochemistry is generally completely negative in SDHB-mutated tumors, whereas SDHD-mutated tumors show residual staining [51]. This suggests that many SDHB variants abolish SDH assembly, while SDHD variants often allow residual SDH assembly. We therefore suggest that SDHB null alleles (consisting of both truncating and missense variants) are more likely to congregate at the low ROS/high succinate end of the spectrum (see Figure 4) compared to many SDHD missense variants. In other words, SDHB null alleles are significantly less likely to cause the high ROS levels which, we postulate, kill sympathetic precursor cells and thus prevent PPGL development (scenario 1), hence explaining the strong association of PPGLs with null alleles. Equally, null alleles may be more likely to cause levels of succinate accumulation high enough to reach the tumorigenic threshold required for the initiation of PPGL development as postulated in scenario 2.

## 5. SDHAF2

Any hypothesis seeking to explain the link between SDH genotype and phenotype must also account for the finding that SDHAF2 variants (of any type) are associated solely with HNPGL [52]. Assembly of the succinate dehydrogenase complex requires at least four dedicated assembly factors, but the SDHAF2 assembly factor is the only subunit currently associated with paragangliomas [53]. SDHAF2 promotes covalent binding of the flavin adenine dinucleotide (FAD) cofactor to SDHA, which is essential for oxidation of succinate [54]. Judged on the basis of function, one might expect that SDHAF2 variants would lead to clinical outcomes similar to those associated with SDHA or SDHB variants. However, all pathogenic variants identified to date lead exclusively to a parasympathetic head and neck paraganglioma phenotype that is also frequent amongst carriers of SDHC and SDHD missense variants [53,55,56,57,58,59,60]. Indeed, of the 40 affected members of the Dutch PGL2 family in which SDHAF2 p.(Gly78Arg) was originally identified, none have developed PPGL during 30 years of surveillance.

Certain aspects of SDHAF2 biology have been addressed [61,62,63] but none of these studies offered a clear explanation of how SDHAF2 dysfunction relates to variants in other complex subunits or explained the phenotype associated with SDHAF2 variants. In light of our proposal that SDHC/D missense variants lead to a high-ROS cellular environment, we suggest a similar mechanism for SDHAF2 but perhaps with even higher ROS levels. The immediate problem with this proposal is the loss of covalent FAD due to SDHAF2 dysfunction. The covalent flavin bond is thought to raise the redox potential of the FAD prosthetic group to enable the enzyme to act as a succinate dehydrogenase [64]. An enzyme lacking covalent FAD is therefore unlikely to have sufficient redox potential to oxidize succinate to fumarate, leading in turn to a lack of succinate-derived electrons necessary for the proposed ROS generation. How then might the SDH complex generate ROS in the absence of covalent FAD? That succinate dehydrogenase can act as a fumarate reductase has long been suggested, especially when FAD is in a non-covalent configuration [65]. In fact, most flavoenzymes harbor FAD in this configuration [64]. Although under normal cellular conditions, reversal of mammalian SDH is thermodynamically unfavorable, the standard reduction potential of UQ is only slightly higher than fumarate, suggesting that net reversal of SDH might occur under certain conditions. Indeed, hypoxia or inhibition of complex III by antimycin was recently shown to lead to UQH2 accumulation and fumarate reduction, reversing SDH activity and electron flow [66]. This process ensures continuing electron transport from complex I and UQ-dependent enzymes such as DHODH, allowing NADH reoxidation and maintenance of the redox balance. Sustained net reversal of SDH was only seen in certain tissues however, including brain and kidney, raising the possibility that fumarate reduction via SDH may be a component of normal biochemistry in neuronal tissues. Furthermore, in a study of eye metabolism, Bisbach and colleagues showed that the hypoxic niche environment of the retina leads to the reduction in fumarate to succinate, followed by the export of succinate to adjoining eyecup tissue, which in turn exported malate that was taken up by retinal cells [67]. These findings accord with the possibility that an SDH complex lacking covalent FAD due to SDHAF2 mutation may act as a fumarate reductase. No study to date has investigated this possibility. Whether SDH-mediated fumarate reductase activity leads in turn to elevated ROS levels or very high levels of succinate remains to be investigated.

From a slightly different perspective, work by Van Vranken et al. indirectly suggested that SDHAF2 may cause SDHA dysfunction [68]. SDHA is the only stable component of the SDH complex, as other subunits are rapidly degraded in the absence of an assembled complex. Van Vranken et al. found that overexpression of yeast Sdh1 (ortholog of SDHA) was toxic in both WT cells and Sdh8 knockout cells, the latter being the ortholog of the SDHAF4 assembly factor. Sdh8 interacts specifically with the flavinated form of Sdh1, so is dependent on flavination by Sdh5/SDHAF2. Further experiments showed that balanced expression of Sdh8 and Sdh1 reduced Sdh1 toxicity in WT cells, suggesting that Sdh8 not only promotes assembly of the Sdh1-Sdh2 dimer (SDHA-SDHB) but also prevents toxicity due to free Sdh1. Various experiments and other lines of evidence pointed to increased oxidative stress as the mechanism underlying Sdh1 toxicity. These authors suggested that tissue dysfunction induced by a loss of SDH may be largely driven by increased ROS production, consistent with the idea that SDH assembly factors may have evolved as a mechanism to protect against ROS production during SDH assembly [68]. These data clearly suggest that mutated SDHAF2 prevents the formation of an SDHA-SDHAF2-SDHAF4 complex, likely resulting in free SDHA and ROS toxicity, which in turn may be preferentially toxic in sympathetic precursor tissues, explaining the exclusive HNPGL phenotype of affected SDHAF2 variant carriers.

Interestingly, a subsequent study by Bezawork-Geleta produced credible evidence supporting a ‘succinate threshold’ hypothesis in relation to SDHAF2 [69]. In this study, the authors examined the role of SDHA following knockout of SDHB. As above, SDHA was found in complex with SDHAF2 and SDHAF4, termed CII_low_ in this study. Various assays suggested that additional knockout of SDHA further reduces cellular fitness compared to SDHB KO, indicating that SDHA in complex with SDHAF2 and SDHAF4 has unexpected biological significance. Perhaps most importantly, loss of CII_low_ led to an increase in NADH, suggesting compromised redox balance and, most surprisingly, also led to reduced levels of succinate and lactate compared to SDHB KO alone. In the context of clinical SDHAF2 variants, this implies that succinate accumulation may fail to reach the tumorigenic threshold, explaining the lack of PPGL cases amongst carriers (Figure 6).

However, what occurs with other possible mechanisms? The functional impact of SDH variants on the primary energy-generating pathway of the cell, the TCA/Krebs cycle, has not been discussed here. Although major disruption of this cycle was a natural predicted outcome once loss of SDHD was identified as a cause of paraganglioma [70], interest in the cellular biochemistry of PPGL/HNPGL tumors has gathered pace only relatively recently [14,15,49,50,71,72]. How specific SDH variant types might result in different biochemical outcomes is presently unclear and a clear link between SDH variants, energy metabolism and tumorigenesis has yet to be established.

## 6. Conclusions

We present here two possible but not mutually exclusive mechanisms by which SDH variants may initiate tumorigenesis. We favor the ROS hypothesis partly because, if proven, the sensitivity of PPGL to ROS may present new therapeutic opportunities [73,74]. We attribute the specific association of SDHB variants with PPGL and malignancy to their limited ability to generate ROS overload (and/or ability to cause excessive succinate accumulation), allowing the emergence of tumors in sympathetic precursor tissues. Functionally, SDHC/SDHD missense variants are primarily associated with HNPGL due to a tissue-specific resistance to ROS and/or succinate levels too low to cause tumors in sympathetic tissues. We argue, and as our previous studies suggest, that HNPGL tissue is relatively agnostic regarding the type of initiating SDH variant. At the clinical level, missense variants seemingly predominate in HNPGL compared to other variants due to the elevated penetrance of SDHD variants and the still unexplained elevated incidence of missense variants as founders. The latter issue is likely to be a stochastic population effect as many types of founder variants have been reported, including truncating variants of SDHB. Collectively, these factors likely explain the apparently specific association of SDHD variants with HNPGL. Are these findings relevant to other variant-specific phenotypes? We strongly suspect, and currently hypothesize in this paper, that the difference between the clinical impact of missense versus truncating variants often results from the interplay between the exact function–dysfunction of a particular protein and a specific tissue. In our opinion, this is likely to be highly specific for each protein–tissue pair and will need to be unraveled in each individual case. Finally, although speculative, the hypotheses presented here are the result of the novel integration of clinical and biochemical knowledge to produce new perspectives. Another important merit is that the many predictions made here are eminently falsifiable.

## Figures and Tables

**Figure 1 genes-13-01025-f001:**
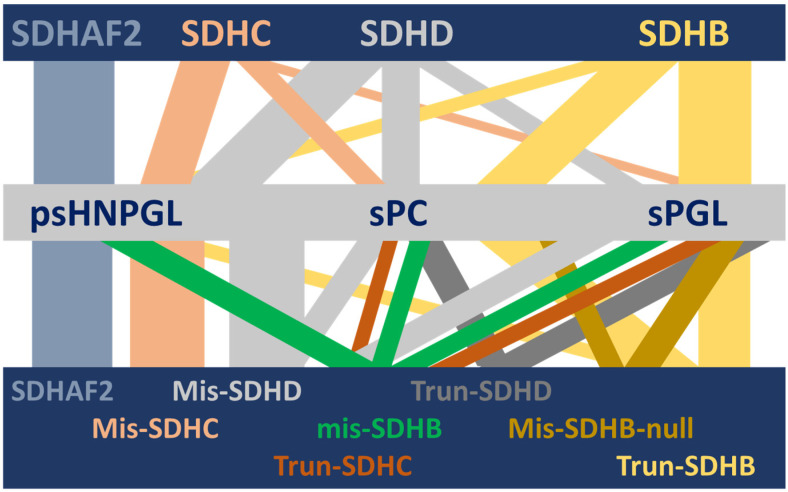
Visual depiction of the complex HNPGL/PPGL clinical spectrum associated with succinate dehydrogenase and its variants [8,9]. Although succinate dehydrogenase is a single protein complex, genetic variants in SDH genes and one assembly factor gene are associated with a range of different but often overlapping HNPGL/PPGL phenotypes. This diversity is unexplained to date. In this illustration, the horizontal bars depict (from top to bottom) the relevant SDH genes, the associated clinical phenotypes and the relevant gene-variant types (SDHAF2 is unannotated in the lower bar because all variant types result in exclusively HNPGL). Vertical bars depict the diversity of phenotypic associations, with the width approximating the strength of a gene–clinical relationship. SDHAF2, SDHC and SDHD show psHNPGL-biased phenotypes, while SDHB is biased towards sympathetic sPC and sPGL tumors. SDHAF2 = succinate dehydrogenase assembly factor 2; SDHC, SDHD, SDHB = succinate dehydrogenase subunit C, D and B, respectively; psHNPGL = parasympathetic head and neck paraganglioma; sPC = sympathetic pheochromocytoma; sPGL = sympathetic paraganglioma; Mis = missense; Trun = truncating; ‘null’ indicates missense variants that act as null alleles.

**Figure 2 genes-13-01025-f002:**
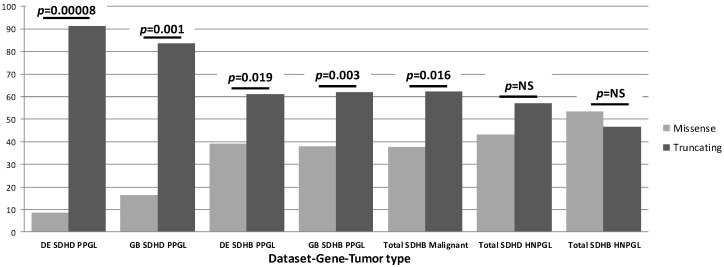
Summary of our SDH genotype-phenotype studies. While all HNPGL data showed an approximately equal distribution of missense and truncating variants, this distribution differed markedly for PPGL cases. The noted differences between variants and the divergent impact across tissues imply distinct biology in both cases. It is important to note that ‘percentage of cases’ refers only to relative numbers of cases attributable to either truncating or missense variants per gene-tumor type and should not be compared across groups. SDHAF2 is not included in this figure or in the relevant studies due to a lack of clinical diversity, as all known clinical cases exhibit solely parasympathetic head and neck paraganglioma.

**Figure 3 genes-13-01025-f003:**
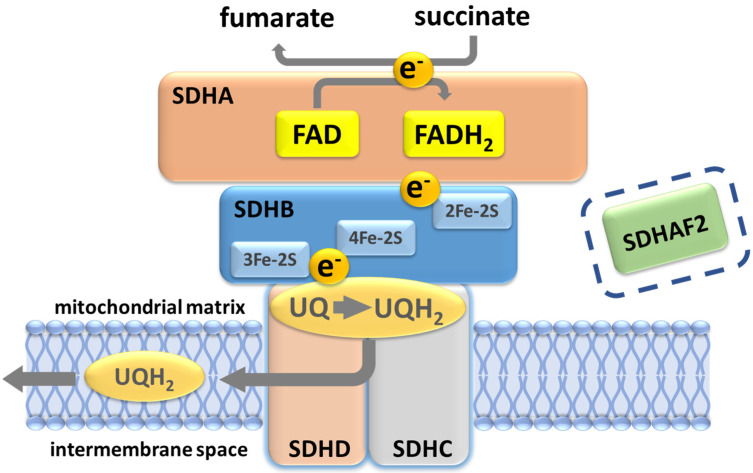
Succinate dehydrogenase/complex II. The oxidation of succinate to fumarate yields two electrons that subsequently reduce ubiquinone to ubiquinol. A ubiquinone binding site (UQ) at the inner mitochondrial membrane, formed by residues in SDHB, SDHC and SDHD, lies close to the SDHB 3Fe-2S redox center. Protonation of ubiquinol likely occurs at the conserved SDHD Tyr114 residue in the UQ pocket, the site of the Northern Italian founder variant p.(Tyr114Cys), which is associated with very low PPGL risk. SDHAF2 is essential for flavination and assembly of the SDH complex and prior to assembly binds at the SDHA-SDHB interface. Although depicted in the figure with regard to the discussion later in this paper, SDHAF2 has no known post-assembly role in SDH function.

**Figure 4 genes-13-01025-f004:**
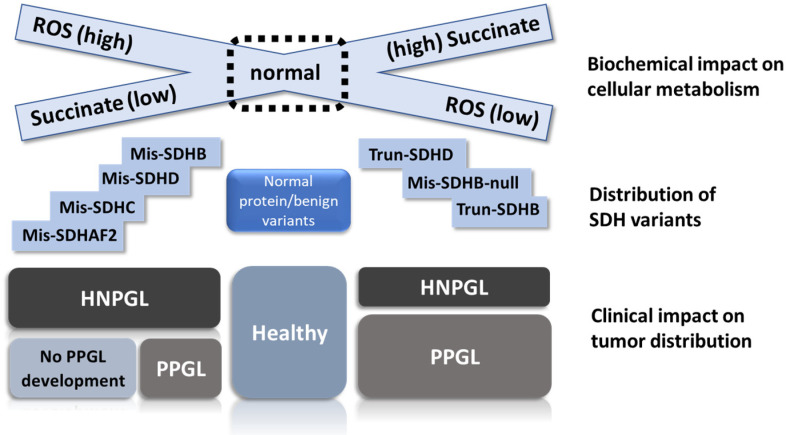
Visual depiction of the relation between SDH biochemistry, SDHx variant type and disease outcome. We propose that ROS accumulation occurs primarily due to the missense variants on the right of the illustration. In this scenario, SDHB missense variants fall primarily into one of two different sets, either as high-ROS hypomorphic alleles similar to SDHC/D missense variants or as low-ROS null alleles functionally equivalent to truncating variants. Truncating variants prevent SDH assembly, leading to the loss of succinate oxidation and accumulation of high levels of the substrate, together with elevated ROS levels that are generally lower compared to missense variants. High levels of succinate limit the generation of ROS at SDHA FAD. HNPGL precursor cells appear relatively agnostic regarding missense or truncating variants, whereas PPGL precursor cells show a substantial bias against SDHAF2, SDHC and SDHD missense variants, likely due to ROS overload and subsequent cell death, resulting in substantially less tumor development compared to better-protected HNPGL precursor tissues. Mis = missense, Trun = truncating, ‘null’ indicates missense variants that act as null alleles.

**Figure 5 genes-13-01025-f005:**
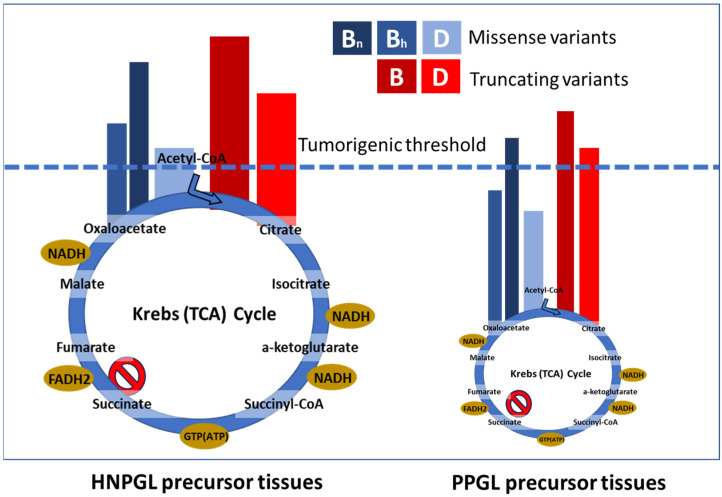
Illustration of scenario 2, the ‘succinate threshold’ model. We propose that succinate accumulation differs markedly between hypomorphic (missense) and null (truncating) alleles. The high succinate levels intrinsic to parasympathetic HNPGL tissues raise cells closer to the tumorigenic threshold compared to sympathetic PPGL precursor tissues. The size of the depicted Krebs cycle approximates the difference in metabolic activity between tissue types. While most SDHC and SDHD variants appear to fall clearly into the expected functional class (with some variation), we predict that many SDHB missense variants are null alleles functionally equivalent to truncating variants (B, SDHB; Bh, SDHB hypomorphic; Bn, SDHB null; D, SDHD).

**Figure 6 genes-13-01025-f006:**
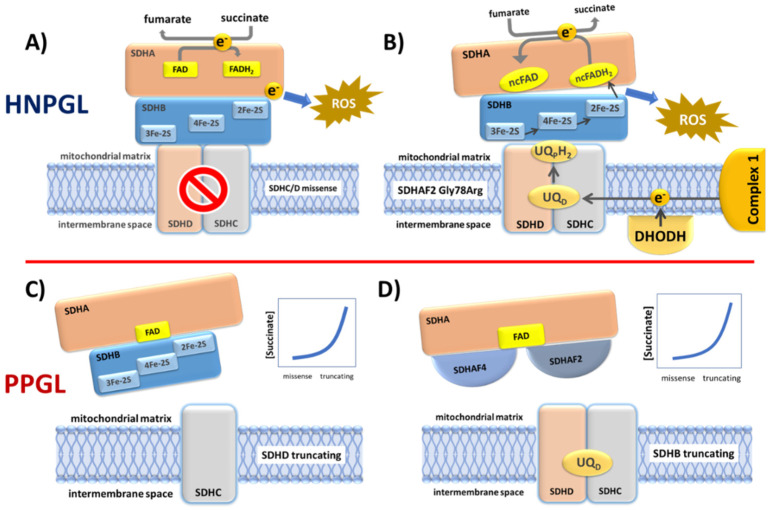
Four possible functional scenarios corresponding to the diverse SDH variant types. (**A**) Many SDHC and SDHD missense variants likely cause reduced rather than total loss of SDH function. When these variants disrupt co-factors involved in electron transport, the result is electron diversion to ROS formation. (**B**) The SDHAF2 Dutch founder variant p.(Gly78Arg) almost completely abolishes covalent flavination of SDHA and is exclusively associated with HNPGL. An SDHA FAD covalent bond is essential to the oxidation of succinate to fumarate. A non-covalent FAD (ncFAD) may enhance the opposite reaction, reduction of fumarate to succinate, producing very high ROS levels. (**C**) SDHC and SDHD truncating variants prevent the assembly of SDH/complex II, resulting in the accumulation of high levels of succinate which likely initiates tumorigenesis via inhibition of 2-oxoglutarate-dependent oxygenases. (**D**) Similarly, the absence of SDHB likely results in a relatively stable SDHA-SDHAF2-SDHAF4 complex which is unlikely to be functional in terms of succinate oxidation or ROS production, again resulting in succinate accumulation.

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
