# Peer review of "Hypothesis: Why Different Types of SDH Gene Variants Cause Divergent Tumor Phenotypes"

_genes, 2022, doi:10.3390/genes13061025_

Round 1
Reviewer 1 Report
In this paper, the authors propose a hypothesis linking aberrations in complex II subunits and development of different cancer phenotypes, and they focus on pheochromocytomas and paragangliomas. This is an interesting paper that provides some new ideas that may be verified, potentially, experimentally. As such, it is pretty good and innovative. However, there are a few drawbacks that ought to be resolved.
One major point concerns the figures. It is unclear what Figure 1 means. There are a few abbreviations scattered over the area. While the explanation is in the legend below, the meaning of the figure is unclear to me, plus it is not very informative. Figure 2 is OK, although it is unclear to me why the assembly factor SDHAF2, which is in other figures, is not included. Figure 3 is completely wrong! First, I do not understand, where the authors got the distal Q-binding site. It is not really defined well and is, rather, putative. Second, fully assembled CII comprises SDHA, SDHB, SDHC and SDHD subunits, but not the assembly factor SDHAF2. Second, SDHAF2 binds to SDHA and flavinylates the catalytic subunit SDHA BEFORE SDHB associates with it. Figure 4 resembles to me, in terms in low level of informativeness, Figure 1. I do not really understand its meaning. Figure 5 seems to be informative, however it is hard to understand without really giving it some hard thinking.#
The authors will know that aberrations in CII are very frequent in renal cancer. In fact, some recent literature show that more than 50%^ patients have non-standard levels of some of the CII subunits and assembly factors. Can the authors also include in their hypothetical work renal cancer?
Author Response
Reviewer 1.
(x) English language and style are fine/minor spell check required
Reply to reviewer: We have done our best but can’t locate spelling errors?
Comments and Suggestions for Authors
General reply to reviewer 1: As so often with reviews, the reviewer provides a valuable fresh perspective on work that has been the sole interest of the authors and a few colleagues for many months.
In this paper, the authors propose a hypothesis linking aberrations in complex II subunits and development of different cancer phenotypes, and they focus on pheochromocytomas and paragangliomas. This is an interesting paper that provides some new ideas that may be verified, potentially, experimentally. As such, it is pretty good and innovative. However, there are a few drawbacks that ought to be resolved.
Reply to reviewer: Thanks for this generous opinion. We are aware that this is a first attempt to feel our way through a dark forest, using the few clues available to locate the correct pathway. We hope that this hypothesis will inform and spur others to reflect on the conundrum of the role of SDH in disease. The models presented by us may indeed prove to be incorrect, but we are confident that much will be learnt during their falsification, bringing light where there is now only darkness.
Reply to reviewer
One major point concerns the figures. It is unclear what Figure 1 means. There are a few abbreviations scattered over the area. While the explanation is in the legend below, the meaning of the figure is unclear to me, plus it is not very informative.
Reply to reviewer: The figures provided with this paper were an (apparently unsuccessful) attempt to provide brief visual summaries of relatively convoluted material, an addition particularly advocated by our clinical colleagues for whom the biochemical arguments are less immediately accessible.
In figure 1 we attempted to convey the idea of a “spectrum of disease”, but that concept was perhaps flawed as the reviewer seems to suggest. We have now adapted the figure to convey the idea of disease complexity originating from dysfunction of a single protein complex with a single function. This conundrum has remained largely unaddressed during the previous 20 years of research. As our earlier work and this paper emphasize, that is in fact a considerable simplification of reality, and a major goal of the present paper is to bring this point to wider attention.
Figure 2 is OK, although it is unclear to me why the assembly factor SDHAF2, which is in other figures, is not included.
Reply to reviewer: We have now addressed this question by adding the following sentence to the figure legend: “SDHAF2 is not included in this figure or in the relevant studies due to a lack of clinical diversity, as all known clinical cases exhibiting solely parasympathetic head and neck paraganglioma (psHNPGL).” To further avoid confusion, we have adjusted our perhaps obscure earlier phrasing in the SDHAF2 section to “all pathogenic variants identified to date (of SDHAF2) lead exclusively to a parasympathetic head and neck paraganglioma phenotype common amongst carriers of SDHC and SDHD missense variants”’.
Figure 3 is completely wrong! First, I do not understand, where the authors got the distal Q-binding site. It is not really defined well and is, rather, putative. Second, fully assembled CII comprises SDHA, SDHB, SDHC and SDHD subunits, but not the assembly factor SDHAF2. Second, SDHAF2 binds to SDHA and flavinylates the catalytic subunit SDHA BEFORE SDHB associates with it.
Reply to reviewer: Figure 3 is a relatively accurate figure in our opinion, so “completely wrong” seems a little harsh. The distal Q-binding site is discussed/described by Silkin et al. Biochimica et Biophysica Acta 1767 (2007) 143–150 and Oyedotun and Lemire, The Journal of Biological Chemistry Vol. 276, No. 20, pp. 16936–16943, 2001. We recognize that it might be controversial but included it for completeness. As it is of no relevance to our arguments in the paper, we are happy to remove it.
We are, of course, aware that SDHAF2 is not a component of assembled SDH and did not intend to depict it as such in the original figure. However, the form and proximity of SDHAF2 as depicted might indeed be interpreted as implying a functional interaction with the assembled SDH complex. It was illustrated based on its relevance to our arguments rather than a biological function in the proximity of the assembled SDH complex. To avoid confusion, we now depict it as yet more detached, isolated by a dotted box and describe its post-assembly role in the figure legend: “Although depicted in the figure with regard to the discussion later in this paper, SDHAF2 has no known post-assembly role in SDH function.” Hopefully this will avoid possible confusion amongst prospective readers.
We are indeed aware that “ SDHAF2 binds to SDHA and flavinylates the catalytic subunit SDHA BEFORE SDHB associates with it.” We were not aware that we have stated the opposite anywhere in this paper. A careful re-reading also supported this view. Perhaps this point was related to our confusing depiction of SDHAF2 in the figure, as discussed above. If not, perhaps the reviewer could elaborate on this point?
Figure 4 resembles to me, in terms in low level of informativeness, Figure 1. I do not really understand its meaning.
Reply to reviewer: We sympathize with the reviewer, having struggled through many iterations to conceive a simple illustration that encapsulates our somewhat complex hypotheses. This figure was intended to convey the essence of our arguments regarding the respective roles of missense versus truncating SDH variants in relation to the proposed biochemical phenotypes (ROS vs. succinate accumulation) and their clinical relationship. Perhaps with careful reading of the appropriate sections, this figure becomes more understandable? Or perhaps not? We are loath to delete it, considering the effort expended.
Figure 5 seems to be informative, however it is hard to understand without really giving it some hard thinking.
Reply to reviewer: In that case we have achieved our objective. This paper was partly inspired by conversations with colleagues in both basic research and clinical care/research. On the one side we have often noted a lack of awareness of basic genetic-clinical correlates, while on the other the necessary background to understand the biological subtleties of SDH function is often lacking. We are convinced that the key to understanding SDH function and the concomitant design of appropriate experiments requires that we bridge these two worlds. We hope in this paper to assist others in achieving this. Illustrations, perhaps from more talented and creative originators than ourselves, can assist in understanding relatively complex material.
The authors will know that aberrations in CII are very frequent in renal cancer. In fact, some recent literature show that more than 50%^ patients have non-standard levels of some of the CII subunits and assembly factors. Can the authors also include in their hypothetical work renal cancer?
Reply to reviewer: We appreciate the comment and agree that the role of SDH in renal cancer is a fascinating one, and indeed may eventually help shed light on its role in paraganglioma-pheochromocytoma. However, while we have three decades of experience in the latter field, our knowledge of the former is limited and does not qualify us to pursue this even in the hypothetical sense. We hope that our present work will inspire others to reflection and we will be happy to read a future paper by the reviewer or another colleague on this subject.
Reviewer 2 Report
In this review the authors try to find scientific explanation of different paraganglioma-pheochromocytoma phenotypes caused by SDHx missense and frame shift/truncating variants. They suggest two metabolic scenarios (ROS model and succinate threshold model) depending on cell type and variant type. The authors provided a detailed and thorough work and the pathobiochemical pathways are nicely summarized, there are several weak and missing points that shoud be corrected in the manuscript:
Major:
- Clinical relevance is almost completely missing
- While phenotype-genotype correlation are frequently referred, several important facts are missing in this field: please see and incorporate:
10.1056/NEJMra1806651
10.1530/ERC-18-0085
10.1002/humu.21136
10.1136/jmedgenet-2017-105127
10.1007/s00432-017-2355-0
- Only PPGL/HNPGL are mentioned regarding SDHx pathogenic variants, however molecular background of other SDH-related tumors would be essential to include when tissue-specific effects are discussed. Please see:
10.3389/fendo.2021.680609
10.3390/medicina56110561
- While behind PPGL/HNPGL development two pathomechanisms are discussed in detail, other SDH-related processes should be summarized as well. Please see:
10.1186/s40170-019-0197-8
10.3390/cancers12113237
10.3390/cancers12030599
- Also, therapeutic relevance should be also be more detailed, including
10.3390/ijms23031450
10.1016/j.bbcan.2018.05.002
Minor:
line 39-41. please revise the sentence as in pheochromocytoma not essentially SDHB and SDHD genes are the most important.
Author Response
Reviewer 2.
Comments and Suggestions for Authors
In this review the authors try to find scientific explanation of different paraganglioma-pheochromocytoma phenotypes caused by SDHx missense and frame shift/truncating variants. They suggest two metabolic scenarios (ROS model and succinate threshold model) depending on cell type and variant type. The authors provided a detailed and thorough work and the pathobiochemical pathways are nicely summarized, there are several weak and missing points that shoud be corrected in the manuscript:
Reply to reviewer:
We thank the reviewer for these positive comments and hope to address some of the suggested weaknesses.
Major:
- Clinical relevance is almost completely missing
Reply to reviewer: This paper describes biochemical hypotheses that seek to explain our recent novel findings concerning SDH variant phenotypes. The clinical correlates of these findings are mentioned but we focus here on the underlying biochemical hypotheses. The broader potential clinical implications of our findings have been previously discussed in our cited papers and will be elaborated in a new worldwide study initiated by us that is currently in the data gathering phase. As this article is already almost 3500 words and thus more than sufficiently taxing for a hypothesis paper, further clinical elaboration, which necessarily remains highly speculative until one or other (or none) of our hypotheses are proven correct, seems to us unnecessary and unproductive at this juncture.
- While phenotype-genotype correlation are frequently referred, several important facts are missing in this field: please see and incorporate:
Reply to reviewer: On reflection the reviewer has highlighted some papers that deserve citation.
The first reference mentioned by the reviewer (10.1056/NEJMra1806651) is indeed relevant and is our reference 1 (see reference list). The second reference, on which one of us was a prominent co-author (10.1530/ERC-18-0085), is an extended summary of the field that is beyond the scope of this article in our opinion and adds little of substance not covered in reference 1.
The third and fourth references (10.1002/humu.21136; 10.1136/jmedgenet-2017-105127) are familiar to us and these important papers have been discussed in our earlier work. On reflection, perhaps it was an oversight to exclude these references from this paper, as this might create the mistaken impression of a claim to exaggerated novelty for our earlier findings. Although our findings are indeed entirely novel, they are not by any means the first attempt to address SDH genotype-phenotype correlations and we would sincerely hope to avoid creating that impression. The references, plus an additional important paper, have now been added to the text with supporting commentary (see line 60 onwards).
The final reference (10.1007/s00432-017-2355-0) is a small study (37 patients) that doesn’t appear to add anything of substance in the present context beyond that of reference 1.
- Only PPGL/HNPGL are mentioned regarding SDHx pathogenic variants, however molecular background of other SDH-related tumors would be essential to include when tissue-specific effects are discussed. Please see: 10.3389/fendo.2021.680609; 10.3390/medicina56110561
Reply to reviewer: We appreciate the comment and agree that the role of SDH in other cancers is a fascinating one, and indeed may eventually help shed light on its role in paraganglioma-pheochromocytoma. However, as we replied to a similar comment by reviewer 1 in the context of renal cancer, with three decades of experience in paraganglioma research we feel adequately qualified to discuss that subject. In contrast, we have no special knowledge of the syndromes/tumors described in the cited references and therefore do not consider ourselves qualified to extend the present hypothesis to those tumors. Should our speculations seem relevant to those with greater expertise we will be happy to read a future paper by others on this subject.
- While behind PPGL/HNPGL development two pathomechanisms are discussed in detail, other SDH-related processes should be summarized as well. Please see: 10.1186/s40170-019-0197-8; 10.3390/cancers12113237; 10.3390/cancers12030599
Reply to reviewer: We are aware that numerous downstream pathomechanisms have been proposed over the last two decades, many of which have isolated but fervent advocates. It is indeed perfectly possible that one of these alternative pathomechanisms is driving PGL tumorigenesis. However, the subject of this paper is not the downstream mechanisms driving tumorigenesis but the detailed mechanisms by which SDH variants initiate tumorigenesis. Indeed, we use the words ‘initiate’ or ‘initiation’ 16 times in the paper. We therefore describe the most compelling/popular pathomechanisms only in passing. Were we to include other pathomechanisms, such as those suggested by the reviewer or the many others, we would need to add an additional section and would risk transforming a hypothesis paper into a review paper, which we sincerely hope to avoid.
We state our focus on initiation of tumorigenesis clearly in the section entitled ‘Mechanisms of tumorigenesis’(line 99 onwards): “We will not speculate on the underlying drivers of downstream tumorigenesis here. Rather, we will argue that certain SDH variants either prevent or cause tumor initiation in specific precursor cell types, building on the premise that specific tissues have higher or lower tolerances to the metabolic disruptions caused by SDH variants.”
It is currently biochemically inconceivable that the primary mechanisms discussed by us do not play a role in the initiation of SDH tumorigenesis (see section ‘Mechanisms of tumorigenesis’, (line 87 onwards) where we mention the accumulation of ‘oncometabolite’ succinate [10-15], 2) the generation of high levels of reactive oxygen species (ROS) [16-20], or 3) via energetic changes due to alterations in SDH/SQR activity [21]). These are the known functions and dysfunctions of the SDH complex. While one should never say never, it seems highly unlikely after many decades of research that an entirely novel activity of the SDH complex will be discovered with relevance to PGL tumorigenesis.
- Also, therapeutic relevance should be also be more detailed, including 10.3390/ijms23031450;
10.1016/j.bbcan.2018.05.002.
Reply to reviewer: As discussed above, these references describe reviews that are no doubt worthy but again are beyond the scope of this ‘functional biochemical’ hypothesis paper. We had no intention to write a review on all aspects of paraganglioma. At this moment the therapeutic relevance, if any, of our hypothesis is unclear and any elaboration would be pure speculation at the expense of the clarity of this paper. We certainly hope to address this issue in the future, should the hypotheses posited here prove to have substance.
Minor:
line 39-41. please revise the sentence as in pheochromocytoma not essentially SDHB and SDHD genes are the most important.
Reply to reviewer: The reviewer is correct. We overlooked the syndromic cases of pheochromocytoma. We have now corrected the sentence to: “Among the most clinically important genes...”.
Reviewer 3 Report
In the manuscript titled “Hypothesis: Why different types of SDH gene variants cause 2 divergent tumor phenotypes” prepared by Bayley and Deville, the author discussed the genetics of SDH-related neuroendocrine tumors. Further, the author discussed possible mechanisms of SDH mutations and ROS productivity. I think overall the manuscript is well-prepared and informative. There is one minor issue that I believe could further improve this work:
The author introduced an important concept that the truncating SDH variants are related to more aggressive phenotypes in the tumor, such as disease incidence and age of onset. The clinical cases with missense variant, however, are relatively benign. This pattern of genotype-phenotype has been reported in other pseudohypoxia-driven cancers. For example, in VHL-associated hemangioblastoma, the truncate mutations are related with stronger aggressive signatures, such as CD31 expression, vasculature formation, and tumor multiplicity (PMID: 23318261). The missense variants may produce functional tumor suppressor that affect the oncogenic process. This information could be helpful, as it not only supports the present work but also provides possible mechanistic resolutions.
Author Response
Reply to Reviewer 3:
In the manuscript titled “Hypothesis: Why different types of SDH gene variants cause 2 divergent tumor phenotypes” prepared by Bayley and Deville, the author discussed the genetics of SDH-related neuroendocrine tumors. Further, the author discussed possible mechanisms of SDH mutations and ROS productivity. I think overall the manuscript is well-prepared and informative. There is one minor issue that I believe could further improve this work:
Reply to reviewer: Thanks for this generous opinion.
The author introduced an important concept that the truncating SDH variants are related to more aggressive phenotypes in the tumor, such as disease incidence and age of onset. The clinical cases with missense variant, however, are relatively benign. This pattern of genotype-phenotype has been reported in other pseudohypoxia-driven cancers. For example, in VHL-associated hemangioblastoma, the truncate mutations are related with stronger aggressive signatures, such as CD31 expression, vasculature formation, and tumor multiplicity (PMID: 23318261). The missense variants may produce functional tumor suppressor that affect the oncogenic process. This information could be helpful, as it not only supports the present work but also provides possible mechanistic resolutions.
Reply to reviewer: We appreciate the comment and understand the apparent similarity.
Firstly, with three decades of experience in the SDH research field, we feel qualified to discuss this topic. This is not, unfortunately, the case regarding VHL, despite the many similarities.
Secondly, we strongly suspect, and currently hypothesize in this paper, that the difference between the clinical impact of missense versus truncating variants results from the specific interplay between the exact function-dysfunction of a particular protein and a specific tissue. In our opinion, this is likely to be highly specific for each protein-tissue pair. In this paper we indeed argue that SDH dysfunction has a very different impact in sympathetic paraganglia tissues compared to parasympathetic tissues. While it would be interesting to speculate on the similarities between VHL and SDH in this context, in this paper we prefer to focus on the proximal impact of SDH rather than speculate on the distal consequences, or perhaps dilute our message by including cases in which we are not expert.
We have now added a paragraph (line 407 onwards) to further emphasize this point:
“Are these findings relevant to other variant-specific phenotypes? We strongly suspect, and currently hypothesize in this paper, that the difference between the clinical impact of missense versus truncating variants often results from the interplay between the exact function-dysfunction of a particular protein and a specific tissue. In our opinion, this is likely to be highly specific for each protein-tissue pair and will need to be unraveled in each individual case.”
Reviewer 4 Report
Bayley and Devilee have compiled an interesting hypothesis paper on the heterogeneous tumour phenotypes of patients with SDHx mutations, focusing specifically on the difference between truncating and missense variants. None of the presented ideas are established or proven; however, it is important to share these ideas with a wider audience. The manuscript is well written and for the most part easy to follow. I have a few points the authors should consider.
1. Figure 1: What is the purpose of the bar on the bottom of the figure? IF it has a purpose it should be stated more clearly in the legend, otherwise I suggest removing it.
2. Figure 4: I find the term “missing tumors” a bit misleading. Why not say, “no tumor development” or “no tumorigenesis”?
3. Line 170-175: This part could do with a bit more clarity and explanation. You introduce the idea that missense variants have different effects than a complete deletion of SDHB. But as shown by Yang et al. (doi: 10.1096/fj.12-210146) certain missense variants lead to a reduction of half-life and degrade. The same is evident from the lack of immunohistochemical staining in these tumours. At this point, one assumes SDHB is not there, similar to a complete deletion. Do you assume that with missense variants, SDHB is present to a low level and that is why complex formation can take place to a limited extent?
4. Line 221-223: I am not convinced that the absolute levels of succinate in a tissue give any information about metabolic activity or rate (also in Figure 5). Is there any other evidence that would support such a claim that carotid body cells have a higher Krebs cycle rate than adrenal medullary cells? If not please, rephrase the sentence to the extent that this is an assumption that is not proven.
5. There are a number of publication by Richter et al, focussed on measuring succinate and other metabolites in large cohorts of PPGL tumours, e.g. doi: 10.1210/jc.2014-2151 and doi: 10.1038/s41436-018-0106-5. The second publication includes a detailed genetic analysis of included SDHx cases. Is there any evidence from this that would support or oppose your hypothesis?
6. It is not clear to me whether your succinate threshold hypothesis also means that generally HNPGL tumour cells should have higher levels of succinate than PPGLs.
7. Line 263-264: You hypothesise that all SDH variants occupy a spectrum from high ROS/low succinate to high succinate/low ROS. Does this statement also include benign variants (why would they have high ROS) or is this limited to the spectrum of pathogenic variants. Please specify.
ROS levels will also be influences by ROS detoxification mechanisms in the cells, e.g. the glutathione synthesis. Does this play a role in your model?
8. Line 279ff. You postulate that SDHB null variants readily initiate tumour formation compared to SDHD missense variants. Why do SDHB mutations have a much lower penetrance than SDHD? And why don’t we find much more HNPGLs with SDHB null variants (e.g. https://doi.org/10.1530/ERC-21-0359)?
9. I agree with the authors in favouring the ROS hypothesis, especially since in vitro studies (references 68, 69) show that treatment with ascorbic acid can lead to a tipping point of ROS overload resulting in cell death and delay of tumour growth. This should be made clearer in the conclusion (line 388).
Author Response
Reviewer 4:
Bayley and Devilee have compiled an interesting hypothesis paper on the heterogeneous tumour phenotypes of patients with SDHx mutations, focusing specifically on the difference between truncating and missense variants. None of the presented ideas are established or proven; however, it is important to share these ideas with a wider audience. The manuscript is well written and for the most part easy to follow. I have a few points the authors should consider.
General reply to reviewer: This reviewer has managed to poke this paper in its most tender places. Although sometimes a little painful, the paper is far better for it, so thank you!
- Figure 1: What is the purpose of the bar on the bottom of the figure? IF it has a purpose it should be stated more clearly in the legend, otherwise I suggest removing it.
Reply to reviewer. We have now more explicitly defined the relevance of each part of the illustration in the accompanying figure legend.
- Figure 4: I find the term “missing tumors” a bit misleading. Why not say, “no tumor development” or “no tumorigenesis”?
Reply to reviewer. Good point. This has been changed to “no PPGL development”, and the figure has been extensively revised (see below).
- Line 170-175: This part could do with a bit more clarity and explanation. You introduce the idea that missense variants have different effects than a complete deletion of SDHB. But as shown by Yang et al. (doi: 10.1096/fj.12-210146) certain missense variants lead to a reduction of half-life and degrade. The same is evident from the lack of immunohistochemical staining in these tumours. At this point, one assumes SDHB is not there, similar to a complete deletion. Do you assume that with missense variants, SDHB is present to a low level and that is why complex formation can take place to a limited extent?
Reply to reviewer. Good point. We have now adjusted the text (178-184) to discuss this point.
“In general, we predict that certain missense variants found in all SDH subunits, but especially SDHB, could cause virtual or complete loss of Complex II assembly and enzyme activity and thus functionally resemble truncating variants [1]. However, we suspect that across the spectrum of missense variant effects, many variants will show significant levels of residual function sufficient to substantially modify clinical phenotype. This will have to be thoroughly investigated for each individual variant, an undertaking already well underway in other fields such as breast cancer [2 ,3]”
- Line 221-223: I am not convinced that the absolute levels of succinate in a tissue give any information about metabolic activity or rate (also in Figure 5). Is there any other evidence that would support such a claim that carotid body cells have a higher Krebs cycle rate than adrenal medullary cells? If not please, rephrase the sentence to the extent that this is an assumption that is not proven.
Reply to reviewer: The evidence requested by the reviewer can be found in existing and newly added references and the references therein. It has been accepted wisdom for many decades that the carotid body is the most highly oxygenated tissue in the body. This level of oxygenation is likely related to its function as the primary oxygen sensor in mammals. An ‘elevated metabolic activity’ is therefore a relatively safe assumption on our part. Nair et al. state that the carotid body “has a relatively high O2 consumption rate (VO2)”, and estimate the effective Km values for oxygen at around 90 Torr, compared to 0.8 Torr for brain and 2.2 Torr for liver.
A high oxygenation and accompanying high metabolic rate are likely essential for the function of the oxygen sensor, regardless of the exact postulated mechanism of that sensor (which remains to be identified). As Daly et al. state: “All these results strongly suggest that circumstances which alter blood pressure and hence the blood flow through the chemoreceptors may influence their activity. Thus, a lowering of the intrasinusal pressure would presumably diminish the carotid body blood flow thereby producing an asphyxial condition in the chemoreceptors. If this explanation is correct, then it is necessary to postulate that, in the presence of a lowered carotid body blood flow, the metabolism of the chemoreceptors is great enough to lower the P02 and/or to raise the pC02 sufficiently to enable these structures to cause their own stimulation.”
Our main point is that existing evidence suggests that the normal carotid body harbors much higher levels of succinate compared to the adrenal medulla. This is relevant in relation to the inhibition of 2OG-dependent oxygenases, probably the most compelling putative distal pathogenic mechanism described in these tumors to date. As such this is not an “assumption” on our part, but a report of existing scientific evidence derived from independent studies. We are not aware of any publication that challenges this view, although we would be interested to learn if the reviewer knows of one.
- There are a number of publication by Richter et al, focussed on measuring succinate and other metabolites in large cohorts of PPGL tumours, e.g. doi: 10.1210/jc.2014-2151 and doi: 10.1038/s41436-018-0106-5. The second publication includes a detailed genetic analysis of included SDHx Is there any evidence from this that would support or oppose your hypothesis?
Reply to reviewer: Also a good point. However, our initial reserve regarding these studies can be found in the Materials and methods section “Fresh frozen tumor tissue (5–10 mg) was homogenized”. Anyone who has handled numerous paragangliomas and pheochromocytomas will be acutely aware of the fact that these tissues vary enormously in the quantity of chromaffin cells, an issue especially pertinent to the analysis of HNPGLs which frequently have a significant stromal content. This problem was also suggested by the wide variation in chemical content of repeat individual tissue samples from the same tumor (Richter 2018 Figures 2a and 2b) and the acknowledgement on page 711 that histological estimation of tumor content was not previously carried out in samples. We would have liked to see Richter and colleagues use some other measure (specific protein content) to determine and correct for the content of these cells in each sample.
Another issue we have with these studies is the use of the succinate:fumarate ratio as a measure of the accumulation of succinate. This approach was introduced by Pollard et al. in 2005 (Hum.Mol.Genet. 2005 Vol. 14: 2231-2239) but without explanation or justification. As succinate accumulates due to disruption of the TCA cycle, the downstream metabolite fumarate can be expected to either relatively decline or to increase, depending on whether a tissue is able to reverse the TCA cycle. As such the ratio perhaps says less about a variant than the raw succinate value as it is in effect an amplification of signal due to TCA disturbance (see Richter 2014 Figure 2). Indeed, Richter and colleagues show that fumarate varies far more widely between SDHB and SDHC/D, and between PGL and HNP, than succinate itself, which is barely significantly different without the inclusion of a succinate:fumarate ratio. Why fumarate differs is independently fascinating and likely important, but perhaps not directly relevant to our succinate hypothesis or the action of SDH variants.
Taking these caveats into consideration and returning to the actual results of Richter et al. we unfortunately find that the level of detail in these studies is too limited to shed light on the current hypothesis. This is the reason why these studies were not discussed in this paper. We nevertheless hope that in the future a refined version of the Richter methodology can be bought to bear on the current question, perhaps initially in the analysis of a uniform cell system carrying various SDH variants. As Susan is a familiar colleague as well as a collaborator in a joint follow-up study on this subject, this is a real possibility and we thank the reviewer for this reminder!
- It is not clear to me whether your succinate threshold hypothesis also means that generally HNPGL tumour cells should have higher levels of succinate than PPGLs.
Reply to reviewer: Yet another good point. Indeed, we lean towards a higher succinate level in pre-tumorigenic tissues and assume that once tumorigenesis is initiated, succinate levels are so elevated as to be functional in all extant tumors. Our central hypothesis is, after all, about tumor initiation and a tissue-specific absence of tumor initiation rather than later processes of tumor development. We have now added a line to clarify this important point:
“This effect is likely independent of subsequent succinate accumulation in clinically apparent tumors but this point should be clarified.”
- Line 263-264: You hypothesise that all SDH variants occupy a spectrum from high ROS/low succinate to high succinate/low ROS. Does this statement also include benign variants (why would they have high ROS) or is this limited to the spectrum of pathogenic variants. Please specify.
ROS levels will also be influences by ROS detoxification mechanisms in the cells, e.g. the glutathione synthesis. Does this play a role in your model?
Reply to reviewer:
We expect that benign variants will have little or no effect on protein function, so likely occupy the sub-clinical, central portion of the proposed spectrum (i.e. normal levels of ROS and succinate). Here we are interested in pathogenic variants of clinical significance however, so have adjusted the sentence to read: “We therefore hypothesize that all clinically-apparent SDH variants occupy a spectrum from high ROS/low succinate to high succinate/low ROS (Figure 4), with normal protein and sub-clinical benign variants occupying the central region of the spectrum.”
This comment was also a spur to reconsider Figure 4. We have never been very convinced of its value in enlightening readers regarding our hypothesis, a shortcoming commented on an by another reviewer. We have now prepared a new figure to hopefully better illustrate this ‘spectrum’ aspect of our hypothesis.
Regarding the latter point, we state in line 179: “We predict that sympathetic paraganglia have weaker anti-oxidative defenses and/or weaker protection against ROS-induced apoptosis compared to oxygen-sensing para-sympathetic tissues, which likely require tight redox control due to the direct involvement of ROS in oxygen-sensing [4,5].” In an earlier draft of the paper we speculated on the exact proteins/mechanisms involved but as this remains purely theoretical at this point, we removed that text for the sake of brevity.
- Line 279ff. You postulate that SDHB null variants readily initiate tumour formation compared to SDHD missense variants. Why do SDHB mutations have a much lower penetrance than SDHD? And why don’t we find much more HNPGLs with SDHB null variants (e.g. https://doi.org/10.1530/ERC-21-0359)?
Reply to reviewer: Perhaps a little beyond the scope of this paper, but interesting none the less. Whether individual SDH variants differ in penetrance is not currently known, to the best of our knowledge. SDHB variants in general may have a lower penetrance due to genetic factors that must be gained or lost prior to initiation of tumorigenesis in a cell. SDHB tumors indeed show characteristic chromosomal gains and losses. SDHD is thought to be highly penetrant due the apparently absolute requirement for loss of a region of maternal chromosome 11p (which harbors a number of imprinted genes), in addition to LOH of the SDHD wildtype allele on chromosome 11p. Deletion of these genetic factors almost invariably occurs via whole chromosome non-disjunction (a relatively common occurrence in all cells) of chromosome 11, except in the rare cases of maternal inheritance which show chromosomal recombination and somatic loss to the maternal 11p region. This phenomenon, which likely drives both the high penetrance and paternal inheritance of SDHD and SDHAF2 tumor risk, has been described in a range of publications from our center detailing what is now known as the ‘Hensen model’.
As for HNPGLs with SDHB null variants, we have shown previously [6,7] and attempted to argue (line 399) here that HNPGL tissue is relatively agnostic regarding the type of initiating variant. We briefly discussed this point in the conclusion but now elaborate on this point:
“We would argue, and our previous studies suggest, that HNPGL tissue is relatively agnostic regarding the type of initiating SDH variant. At the clinical level, missense variants seemingly predominate in HNPGL compared to other variants due to the elevated penetrance of SDHD variants and the still unexplained elevated incidence of missense variants as founders. The latter issue is likely to be a stochastic population effect as many types of founder variants have been reported, including truncating variants of SDHB. Taken together, these factors likely explain the apparently specific association of SDHD variants with HNPGL.”
- I agree with the authors in favouring the ROS hypothesis, especially since in vitro studies (references 68, 69) show that treatment with ascorbic acid can lead to a tipping point of ROS overload resulting in cell death and delay of tumour growth. This should be made clearer in the conclusion (line 388).
Reply to the reviewer: Quite honestly, we feel that we already heavily favor the ROS hypothesis in the conclusion, perhaps even a little too much, considering the plausibility of the succinate threshold model. We have made a couple of minor alterations to the text but prefer not to go any further.
References
- Yang, C.Z.; Matro, J.C.; Huntoon, K.M.; Ye, D.Y.; Huynh, T.T.; Fliedner, S.M.J.; Breza, J.; Zhuang, Z.P.; Pacak, K. Missense mutations in the human SDHB gene increase protein degradation without altering intrinsic enzymatic function. Faseb Journal 2012, 26, 4506-4516, doi:10.1096/fj.12-210146.
- Guidugli, L.; Carreira, A.; Caputo, S.M.; Ehlen, A.; Galli, A.; Monteiro, A.N.A.; Neuhausen, S.L.; Hansen, T.V.O.; Couch, F.J.; Vreeswijk, M.P.G.; et al. Functional Assays for Analysis of Variants of Uncertain Significance in BRCA2. Human Mutation 2014, 35, 151-164, doi:10.1002/humu.22478.
- Boonen, R.A.C.M.; Rodrigue, A.; Stoepker, C.; Wiegant, W.W.; Vroling, B.; Sharma, M.; Rother, M.B.; Celosse, N.; Vreeswijk, M.P.G.; Couch, F.; et al. Functional analysis of genetic variants in the high-risk breast cancer susceptibility gene PALB2. Nature Communications 2019, 10, doi:ARTN 529610.1038/s41467-019-13194-2.
- Lopez-Barneo, J.; Ortega-Saenz, P. Mitochondrial acute oxygen sensing and signaling. Crit Rev Biochem Mol 2021, doi:10.1080/10409238.2021.2004575.
- Swiderska, A.; Coney, A.M.; Alzahrani, A.A.; Aldossary, H.S.; Batis, N.; Ray, C.J.; Kumar, P.; Holmes, A.P. Mitochondrial Succinate Metabolism and Reactive Oxygen Species Are Important but Not Essential for Eliciting Carotid Body and Ventilatory Responses to Hypoxia in the Rat. Antioxidants-Basel 2021, 10, doi:ARTN 840
10.3390/antiox10060840.
- Bayley, J.P.; Bausch, B.; Jansen, J.C.; Hensen, E.F.; van der Tuin, K.; Corssmit, E.P.M.; Devilee, P.; Neumann, H.P.H. SDHB variant type impacts phenotype and malignancy in pheochromocytoma-paraganglioma. Journal of Medical Genetics 2021, doi:10.1136/jmedgenet-2020-107656.
- Bayley, J.P.; Bausch, B.; Rijken, J.A.; van Hulsteijn, L.T.; Jansen, J.C.; Ascher, D.; Pires, D.E.V.; Hes, F.J.; Hensen, E.F.; Corssmit, E.P.M.; et al. Variant type is associated with disease characteristics in SDHB, SDHC and SDHD-linked phaeochromocytoma-paraganglioma. J Med Genet 2020, 57, 96-103, doi:10.1136/jmedgenet-2019-106214.
Round 2
Reviewer 1 Report
THe manuscript has been improved sufficiently.
Author Response
Do I need to comment on the below?